# Ti/PbO_2_ Electrode Efficiency in Catalytic Chloramphenicol Degradation and Its Effect on Antibiotic Resistance Genes

**DOI:** 10.3390/ijerph192315632

**Published:** 2022-11-24

**Authors:** Hao Liu, Luwei Zhai, Pengqi Wang, Yanfeng Li, Yawei Gu

**Affiliations:** 1Shandong Tiantai Environmental Technology Co., Jinan 250101, China; 2School of Environmental Science and Engineering, Qilu University of Technology, Shandong Academy of Sciences, Jinan 250353, China; 3Henan Key Laboratory of Water Pollution Control and Rehabilitation Technology, Henan University of Urban Construction, Pingdingshan 467036, China

**Keywords:** Ti/PbO_2_ electrodes, electrochemical oxidation, chloramphenicol, degradation effect, antibiotic resistance genes

## Abstract

Livestock farming has led to the rapid accumulation of antibiotic resistance genes in the environment. Chloramphenicol (CAP) was chosen as a model compound to investigate its degradation during electrochemical treatment. Ti/PbO_2_ electrodes were prepared using electrodeposition. The prepared Ti/PbO_2_-La electrodes had a denser surface and a more complete PbO_2_ crystal structure. Ti/PbO_2_-Co electrodes exhibited improved electrochemical catalytic activity and lifetime in practice. The impact of different conditions on the effectiveness of CAP electrochemical degradation was investigated, and the most favorable conditions were identified (current density: I = 15.0 mA/cm, electrolyte concentration: c = 0.125 mol/L, solution pH = 5). Most importantly, we investigated the effects of the different stages of treatment with CAP solutions on the abundance of resistance genes in natural river substrates (*intI1*, *cmlA*, *cmle3*, and *cata2*). When CAP was completely degraded (100% TOC removal), no effect on resistance gene abundance was observed in the river substrate; incomplete CAP degradation significantly increased the absolute abundance of resistance genes. This suggests that when treating solutions with antibiotics, they must be completely degraded (100% TOC removal) before discharge into the environment to reduce secondary pollution. This study provides insights into the deep treatment of wastewater containing antibiotics and assesses the environmental impact of the resulting treated wastewater.

## 1. Introduction

In 1999, Daughton and Ternes concluded that the long-term use of large quantities of pharmaceuticals and personal care products (PPCPs) would lead to serious environmental problems, and they identified PPCPs as a new type of pollutant [1]. More than 50,000 PPCPs are already used in everyday life, and the annual PPCP production is well over 30 million tons [2]. Antibiotics, a common type of PPCP, are inexpensive, readily available, and structurally stable; only 15% of antibiotics are absorbed by the body after ingestion, with the remainder entering the environment in various ways [3,4]. Transport transformation and the efficient removal of PPCPs and antibiotics from aqueous environments have become research hotspots. Conventional wastewater treatment technologies are inefficient at completely degrading antibiotics [5]. Antibiotics continue to accumulate throughout the food chain, posing a potential hazard to human health and the environment [6]. The adsorption technique is advantageous, because it is a simple and environmentally friendly process, but it does not completely degrade pollutants. Carrales-Alvaradoa et al. [7] synthesized nanomaterials from graphene oxide and graphite to treat metronidazole and methicillin antibiotics in water. The results showed that the maximum adsorption of graphene oxide at T = 25 °C was 218 mg/g for meperidine at pH 10 and 190 mg/g for metronidazole at pH 7. Photocatalytic oxidation technology boasts advantages of simple operation, low energy consumption, and no secondary pollution [8,9]. It is mostly used to degrade low concentrations of organic matter. To further improve the performance of TiO_2_ photocatalytic materials, Ioannidou et al. [10] used tungsten oxide to modify TiO_2_, and the degradation under simulated solar irradiation increased the catalytic degradation rate by 50%, compared to the pristine P25-TiO_2_ photocatalytic material. Liu et al. [11] successfully synthesized an efficient z-type SnS2/MIL-88B (Fe) (SnSFe) photocatalyst. Their study showed that SnS_2_/MIL-88B (Fe) (SnSFe) has great potential in wastewater remediation.

The misuse of antibiotics has led to the development of antibiotic-resistant bacteria (ARB) and antibiotic resistance genes (ARGs) [12]. The worldwide spread of ARB and ARGs is positively correlated with the use of antibiotics and is rapidly increasing [13,14]. The misuse of antibiotics causes antibiotic resistance in humans, a serious problem responsible for approximately 700,000 deaths worldwide each year [15]. River ecosystems are the main sites of ARB and ARG storage, and river sediments are especially important reservoirs [16]. The abundance of in vivo and in vitro ARGs in the estuarine sediments of the Haihe River Basin in China is 3.31 × 10^7^–2.93 × 10^8^ copies/g and 9.06 × 10^6^–1.32 × 10^8^ copies/g, respectively. Antibiotic contamination and ARB presence in the environment pose a serious threat to ecology and human health, and antibiotics constitute an emerging pollutant class that needs to be controlled. Therefore, complete antibiotic removal using deep treatment technologies is essential to guarantee ecological safety.

Electrochemically advanced oxidation processes can be used to degrade organic pollutants into CO_2_, H_2_O, and inorganic small molecules by generating highly oxidizing reactive oxygen species, using a green and efficient process, free from secondary pollutants [17,18]. Brinzila et al. [19] used boron-doped diamond electrodes to degrade tetracycline and found that it could be completely degraded after a 4 h reaction, even at high concentrations (>150 mg/L). Sun et al. [20] prepared a Ti-Sn/γ-Al_2_O_3_ three-dimensional particle electrode that achieved a chloramphenicol (CAP) degradation higher than 70% under the optimal process conditions (current strength, 0.1 A; air flow rate, 1.0 L/min; conductivity, 6000 ΜS/cm; pH = 6). The Ti/Ti_4_O_7_ anode prepared by JB Wang et al. [21] could degrade 95.80% of tetracycline within 40 min. Wang et al. [22] used a Ti/SnO_2_-Sb/Er-PbO_2_ electrode for the electrochemical degradation of sulfamethoxazole and acetamethoxazole. After 3 h of reaction, the sulfamethoxazole TOC removal was nearly 63.2%, nitrogen conversion into NO_3_^−^ was 22.4%, and the sulfur conversion was 98.8%. Liang et al. [23] investigated the CAP degradation effect and mechanism using a bioelectrochemical system, which completely removed the antibacterial activity of CAP.

Ti/PbO_2_ electrodes are suitable for conducting electrochemical catalytic reactions and are chemically stable under strong acidic and current conditions [24]. Currently, anodic oxidation is the most widely used method to prepare Ti/PbO_2_ electrodes. During preparation and use, anode oxidation forms a passivation layer (TiO_2_), which reduces the stability and activity of the Ti/PbO_2_ electrode. In addition, Pb^2+^ leaching during electrolysis can result in secondary environmental pollution. To resolve the problems affecting the service life and catalytic activity of Ti/PbO_2_ electrodes, researchers have improved and enhanced their performance by introducing other particles into the active layer of PbO_2_. Experimental results confirm that doping active particles can effectively improve the electrode surface morphology, leading to higher catalytic activity and longer service life. PbO_2_ and SnO_2_ have the same tetragonal crystal system and similar crystal structures. The metal activity of excess elements is second only to that of alkali and alkaline earth metal elements [25]. As highly abundant transition metals, rare earth metal elements have been widely used to modify electrocatalytic materials and improve their electrocatalytic performance. Co, a common transition metal, improves the surface structure of the PbO_2_ active layer and increases its catalytic activity.

In this study, the effect of CAP-treated wastewater at different treatment stages on the abundance of ARGs in river bottom sediments was investigated in addition to conventional structural characterization and performance tests. This has important implications for evaluating the potential ecological impact of the wastewater discharged into the environment at different CAP treatment stages.

## 2. Materials and Methods

### 2.1. Experimental Materials

CAP and ampicillin were purchased from Yuanye Biotechnology Co. (Shanghai, China). SnCl_4_, SbCl_3_, Co(NO_3_)_2_, and Pb(NO_3_)_2_ were acquired from the Shanghai Maclean Biochemical Technology Co. (Shanghai, China). Tryptone, yeast dip powder, and agar powder were obtained from Haibo Biotechnology (Taizhou, China). A rapid plasmid extraction kit, DNA purification and recovery kit, soil genomic DNA extraction kit, 2× Taq Plus PCR premix reagent, pLB zero background rapid cloning kit, TOP10 receptor cells, and the SuperReal fluorescence quantification premix reagent were bought from Tiangen Biochemical Technology Co. (Beijing, China). All other reagents were purchased from Sinopharm Chemical Reagent Co. Ltd. (Shanghai, China).

### 2.2. Experimental Apparatus

The DC voltage regulator (KPS-3005DU) was acquired from Shenzhen Zhaoxin Electronic Instruments Co. (Shenzhen, China). A box-type resistance furnace (SX2-4-10Z) was obtained from Shanghai Boxun Industrial Co. (Shanghai, China). An electrochemical workstation (CHI 660C) was bought from the Shanghai Chenhua Instrument Co. (Shanghai, China). A high-performance liquid chromatography system (LC-20ATvp) was purchased from Shimadzu Corporation (Shimadzu, Japan). A scanning electron microscope (Sigma 300) was acquired from Zeiss (Oberkochen, Germany). An X-ray energy spectrum analyzer (QUANTAX) was obtained from Bruker (Fremont, Germany). XPS (EscaLab Xi+) was obtained from Thermo Fisher Scientific (Waltham, MA, USA). An ultrasonic cleaner (FS-613HT) was purchased from Dongguan Kongshijie Ultrasonic Technology Co. (Dongguan, China). An intelligent light incubator (GXZ) was purchased from Ningbo Jiangnan Instrument Factory (Ningbo, China). A high-speed centrifuge (D2012) and vortexer (MX-S) were acquired from Sero Czech, Stafford, TX, USA. A Nanodrop device (ND-Lite), PCR gene amplification instrument (Miniam), and real-time fluorescence quantitative PCR (qPCR) instrument (7500) were obtained from Thermo Fisher Scientific (Waltham, MA, USA). An electrophoresis instrument (DYY-6C) was obtained from Beijing Liuyi Biotechnology Co. (Beijing, China). A UV analyzer (JY02S) was acquired from Beijing Junyi Oriental Electrophoresis Equipment Co. (Beijing, China). An autoclave sterilizer (MJ-78A) was purchased from Schdukai Instruments Co. (Shanghai, China). A thermostatic incubation shaker (TS-2100B) was acquired from the Shanghai Tiancheng Experimental Instrument Manufacturing Co. (Shanghai, China) and a vacuum freeze-dryer (4.5 L) was obtained from LABCONCO (Kansas City, MI, USA).

### 2.3. Experimental Methods

#### 2.3.1. Ti/PbO_2_ Electrode Preparation

The Ti/PbO_2_-Co electrode preparation process is shown in Figure 1. The Ti plates were sandpapered until smooth and heated in a 10% sodium hydroxide solution in an 80 °C water bath for 3 h. The Ti plates were initially oxidized using a mixture of H_2_SO_4_ and HNO_3_ (1:1). Then, they were immersed in an oxalic acid solution at a mass fraction of 15%, which was boiled for 3 h. Etching was performed on the surface of the Ti substrate to increase the surface adhesion.

The gel was prepared by dissolving 2 mL of hydrochloric acid in 50 mL of isopropanol and then adding 0.2 g of citric acid and stirring, followed by adding 14.02 g of tin chloride pentahydrate and 1.02 g of antimony chloride. The gels were thermally baked (T = 200 °C, t = 30 min) and calcined (T = 550 °C, t = 2.5 h) after the Ti plates were uniformly covered.

The PbO_2_ deposition solution contained 165.6 g/L of Pb(NO_3_)_2_, 37.52 g/L of Cu(NO_3_)_2_, 0.42 g/L of NaF, and 6.30 g/L of HNO_3_. The Co(NO_3_)_2_ concentration was adjusted to 1.2 g/L. The experimental conditions used for electrodeposition included a current density of 10 mA/cm^2^, an electrode spacing of 30 mm, a Cu electrode as the cathode, a reaction time of 4 h, and a temperature of 18 °C.

#### 2.3.2. CAP Test Method

The mobile phase was a water (pH = 2.5, adjusted using H_3_PO_4_):methanol (70:30) solution. The flow rate was 1.5 mL/min, and the injection volume was 20 μL. The chromatographic column was an InertSustain C18 (4.6 × 150 mm, 5 μm) filled with octadecylsilane-bonded silica gel.

#### 2.3.3. High Throughput Sequencing and qPCR

The experimental conditions were as follows: the anode was a Ti/PbO_2_-Co electrode, current density was 15 mA/cm^2^, concentration of the electrolyte Na_2_SO_4_ was 0.125 mol/L, the solution pH was 7, and CAP concentration was 10 mg/L. Samples of 400 mL were collected at different treatment stages and mixed with 600 mL of river water (location: 36.546002 N, 116.802059 E). A 12 h light/dark cycle was used for the incubation to simulate the conditions of a natural environment. Samples were collected on days 0, 5, 20, and 50 to determine the absolute abundance of ARG in the sediment organisms.

The experiments were performed under sterile conditions. Plasmids for each gene were constructed using DNA extraction, PCR amplification, agarose gel electrophoresis, and ligation transformation. A Nanodrop spectrophotometer was used to measure the absorbance values (OD_260_) of the complete and identified plasmids. Individual plasmids were diluted 10-fold (45 µL of the dilution solution + 5 µL of the plasmid). The qPCR reaction system (20 μL) consisted of 10 μL of 2× SuperReal PreMix Plus reagent, 0.6 μL each of the forward and reverse primers, 0.2 μL of 50× ROX reference dye, and 1 μL of DNA template. qPCR was performed using a three-step reaction procedure: a pre-denaturation phase at 95 °C for 15 min (1 cycle), pre-denaturation phase at 95 °C for 10 s, and an extension phase at 72 °C for 32 s (40 cycles). The amplification efficiency obtained was 100% ± 10%, with R^2^ values above 0.99.

### 2.4. Data Analysis

The XRD images of the electrodes were fitted and analyzed using the MDI Jade 6 software. All graphs in the article were plotted using the Origin 2017 software.

## 3. Results and Discussion

### 3.1. Morphological and Structural Characterization

The microscopic morphology and fine structure of the Ti/PbO_2_ electrode surface were observed using SEM at ×500, ×1000, ×2000, and ×4000 magnifications (Figure 2). Magnification at ×500 and ×1000 showed that the electrode surface was covered with PbO_2_ crystals and had a dense active layer. The SnO_2_-Sb_2_O_3_ interlayer [26] and F^−^ [27] in the deposition solution created this denser electrode active layer. Using Co doping, the Ti/PbO_2_ electrode surface became denser, which extended the lifetime of the device and increased its catalytic activity [28]. At ×2000 and ×4000 magnification, the PbO_2_ crystals in the Ti/PbO_2_-Co electrode had an intact cubic cone structure, and those in the Ti/PbO_2_ electrode had a broken cubic cone structure. A higher number of defects on the PbO_2_ crystal surface enabled more Pb^2+^ to enter the solution, diminishing the active layer and affecting the catalytic activity and lifetime of the electrode, and the release of Pb^2+^ to the environment can lead to more serious ecological risks [29].

The main diffraction peaks of the Ti/PbO_2_ electrode matched perfectly with β-PbO_2_ (41-1492), corresponding to the (110), (101), (200), (211), (310), (301), and (202) crystal planes of the β-PbO_2_ crystal (Figure 3). When the electrodeposition solution pH was 1–3, β-PbO_2_ crystalline forms were obtained [30]. Moreover, β-PbO_2_ crystal formation was promoted when the current density was 2–20 mA/cm^2^. The β-PbO_2_ crystals exhibited better electrical conductivity and electrocatalytic properties. Doping the active layer of PbO_2_ with Co induced a greater shift toward the (110) and (101) crystallographic planes. A more exposed high-index surface led to lower OEP [31]. The average grain size of the PbO_2_ crystals in the Ti/PbO_2_ electrode was 62.78 nm, whereas that in the Ti/PbO_2_-Co electrode was 40.61 nm, according to the Debye–Scherrer formula [32].

The elemental peaks of Pb, C, and O are clearly visible in the full spectrum of the Pb electrode (Figure 4a). Co doping increased the intensity of the Pb diffraction peaks. This indicates that Co doping favors PbO_2_ deposition. The diffraction peaks of Pb-O and 2PbCO_3_-Pb(OH)_2_ appeared in the XPS pattern of the Pb electrodes (Figure 4b). This included intermediate products and by-products of the electrodeposition process, confirmed by the findings of Gao et al. [33].

### 3.2. Electrochemical Property Characterization

Doping the PbO_2_ active layer with Co increased the starting potential to 1.63 V and increased the overpotential by 62.68% to 789 mV (Figure 5a). These results indicate that doping the PbO_2_ active layer with Co can inhibit oxygen precipitation and the higher starting potential can reduce the energy consumption of the reaction during pollutant degradation [34].

Ti/PbO_2_ electrodes undergo three stages during use: activation, stabilization, and deactivation, when the voltage exceeds 10 V [35]. Elemental Co doping increased the service life of the electrodes to 204 days, an improvement of 47.82% (Figure 5b,c). During the test, O_2_ was produced on the electrode surface. As the test progressed, O_2_ came into contact with the Ti substrate through the gaps between the active layers, producing an oxide film. Simultaneously, the active layer reacted with H_2_O_2_ in the system to produce PbSO_4_, which intensified the shedding of the active layer. Co doping of the PbO_2_ active layer effectively filled the voids between the PbO_2_ crystals, resulting in a denser structure. La doping of the PbO_2_ active layer not only improved its surface morphology, but also increased the service life of the electrode.

Radicals are less stable and have shorter lifetimes, with ·OH and SO_4_^−^ half-lives of 1 × 10^−9^ and 3 × 10^−5^–4 × 10^−5^ s, respectively [36]. Direct determination of the concentration of these radicals is more difficult, and scavengers usually react with the most reactive radicals, followed by qualitative analysis of the analytical reagents or products. The rates of methanol reaction with ·OH and SO_4_^−^ were 9.7 × 10^8^ and 1.0 × 10^7^ M^−1^s^−1^, respectively. The tert-butanol reaction rates with ·OH and SO_4_^−^ were 3.8 × 10^8^–7.6 × 10^8^ and 4 × 10^5^–9.1 × 10^5^ M^−1^s^−1^, respectively. After 70 min of catalytic degradation, the radical degradation rates were reduced by 29.75% (tert-butanol) and 50.82% (methanol). After its addition to the solution, tert-butanol reacts preferentially with ·OH in the solution, thus preventing it from reacting with the organics. After the addition of methanol to the solution, the oxidizing reactive groups such as ·OH and SO_4_^−^ were completely captured during the reaction due to the similar reactivity of methanol with ·OH and with SO_4_^−^ (Figure 5d). Furthermore, the CAP degradation rate decreased by 50.82% after the addition of methanol. Thus, organic matter was mainly oxidized and decomposed directly on the electrode surface.

### 3.3. Studies Affecting the CAP Degradation Effect

As a transition metal, Co exhibits a high metallic activity [25]. CAP degradation was 99.98% (t = 50 min) and TOC removal was 98.19% (t = 12 h) when using the Ti/PbO_2_-Co electrode, which were 21.28% and 65.86% higher than the rates obtained when using the Ti/PbO_2_ electrode, respectively (Figure 6a,b). Co doping improved the ability of the Ti/PbO_2_ electrode to degrade CAP and remove TOC.

A current density of 15 mA/cm^−2^ was the most favorable for CAP degradation, allowing its complete degradation within 70 min as well as its complete removal within 12 h (Figure 6c,d). This was attributed to the increased production of reactive oxides such as ·OH, which resulted from the increased current density [37]. However, higher current densities (I = 20 mA/cm^2^) did not improve the CAP degradation. The CAP degradation reaction was accompanied by side reactions. When the current density increases, the side reactions compete with CAP degradation and reduce its efficiency [38]. At the same time, high ·OH concentrations are generated per unit time, and ·OH itself undergoes self-quenching reactions to produce O_2_, reducing··OH utilization [39].

The electrolyte was used as a carrier to improve the conductivity and current efficiency [40]. Figure 6e,f shows that 0.125 mol/L was the optimal electrolyte concentration for CAP degradation, reaching 100% within 50 min, and complete CAP mineralization could be achieved within 10 h. Conductivity increases along with the electrolyte concentration in the low electrolyte concentration range, which can increase the electrochemical reaction rate [41]. When the electrolyte concentration exceeds a certain threshold, the SO_4_^2−^ plasma adsorbs onto the electrode surface, reducing the chances of direct CAP oxidation on the anode surface [42]. During electron transfer, sodium sulfate generates persulfate (S_2_O_8_^2−^) and sulfate (SO_4_^−^) radicals with oxidizing properties (Equations (1) and (2)), which are less reactive than ·OH [43].

2SO_4_^2−^→S_2_O_8_^2−^+ 2e^−^,(1)

S_2_O_8_^2−^+ e^−^→SO_4_^2−^+SO_4_^−^,(2)

As shown in Figure 6g,h, CAP (≤5 mg/L) was completely degraded within 40 min. When the catalytic degradation time increased to 70 min, CAP (≤20 mg/L) was completely degraded. At high organic matter concentrations, reactive groups such as ·OH can be involved in organic matter degradation, increasing their utilization [44].

One of the important reference factors in the treatment of organic wastewater via electrochemical oxidation is pH; for example, metronidazole has a better degradation effect in acidic media [45], whereas cloxacin performs better under alkaline conditions [46]. After 50 min of catalytic degradation (Figure 6i,j), the CAP degradation rate was 100% at pH 2 and 86.19% at pH 12. Thus, acidic conditions were more favorable for CAP degradation. ·OH has a higher oxygen precipitation potential (2.85 V) under acidic conditions, which inhibits its decomposition and oxygen precipitation, resulting in more ·OH involved in CAP degradation [47]. Under alkaline conditions, ·OH is converted to the less oxidative ·O^−^, thus reducing the effectiveness of the catalytic degradation [48]. CAP sheds chlorine molecules during degradation, and acidic conditions are conducive to the production of hypochlorous acid, a strong oxidizing substance [49]. In addition, a high pH can lead to the aggregation of organic matter near the anode, which affects the degradation effect while accelerating electrode passivation [34].

### 3.4. Effect of CAP Degradation Intermediates on Resistance Genes

Four ARGs including integrons (*intI1*) and CAP-like resistance genes (*cmlA*, *cmle3*, and *cata2*) were detected in the river substrates.

Mobile genetic elements (MGEs) are important for ARG reproduction and transmission [50]. *IntI1* is located on plasmids and transposons and stimulates ARG expression and their horizontal gene transfer [51,52]. The absolute abundance of *intI1* in the EG720 samples remained unchanged after 50 days of incubation (Figure 7). With increasing incubation time, *intI1* abundance in the CAP-containing and CAP degradation intermediate samples gradually increased, with EG30 showing the highest growth rate. The combination of CAP and its degradation intermediates contributed to an increased *intI1* abundance and may have also contributed to the spread of ARGs. MGEs have been detected in various environmental samples, suggesting that they may be an important factor in exacerbating the public health problem of ARGs [53]. MGEs, one of the most important pathways through which ARGs spread in the environment, critically influence ARG transport and transformation. Therefore, researchers should focus on the abundance of both ARGs and MGEs, which are closely related, to inhibit ARG spread and proliferation.

As shown in Figure 8, when the culture was completed (50 days), the absolute abundance of CAP resistance genes in the EG720 samples changed the least compared to that of the CK group, and remained essentially unchanged, a trend identical to that observed for the absolute abundance of *intI1*. CAP was completely mineralized to CO_2_ and H_2_O after 720 min during the substrate incubation process, leading to a lower impact on microorganisms. Compared to that in the CK group, the abundance of *cmlA*, *cmle3*, and *cata2* genes increased in the EG0, EG30, and EG180 treatment groups, with the greatest increase observed in the EG180 treatment group. As shown in Figure 8, the absolute abundances of *cmlA*, *cmle3*, and *cata2* increased with time. When CAP was not completely degraded (EG0, EG30, and EG180), the remaining CAP and its degradation intermediates stimulated the replication and propagation of CAP resistance genes. As the incubation time increased, the effect on the CAP resistance genes also tended to increase. Thus, the intermediate products of CAP degradation promote ARG expression, namely, integron expression.

In recent years, ARGs have shown a gradual enrichment trend in aquatic environments [54]. With the rapid urban development, a large amount of urban domestic sewage has been discharged into urban water bodies without deep treatment. These effluents contain high concentrations of antibiotics and other PPCPs, which can enrich ARGs in urban water sections once they enter the natural river environment in cities [55]. The overlying water column influences river sediments, which can reflect the pollution status of the sampling site over a longer timescale; however, studies on river sediments are lacking.

## 4. Conclusions

The surface morphology, crystal structure, and electrochemical properties of Ti/PbO_2_-Co electrodes were characterized. After doping the PbO_2_ active layer with Co, the PbO_2_ crystals became more closely arranged, effectively improving the morphology of the Ti/PbO_2_ electrode. Higher β-PbO_2_ crystal catalytic activity and smaller crystal size were obtained at pH 3 and I = 6 mA/cm^2^. The Ti/PbO_2_-Co electrode exhibited a higher onset and overpotential in the LSV and accelerated lifetime tests, and its lifetime increased to 204 days. Free radical trapping experiments confirmed that CAP reacted mainly on the electrode surface. Elemental doping of the PbO_2_ active layer increased the catalytic activity of the Ti/PbO_2_ electrode.

The CAP electrochemical catalytic degradation and TOC removal processes were both primary kinetic reactions. Both reactions increased along with current density; however, higher current densities (I = 20 mA/cm^2^) led to worse catalytic results. Na_2_SO_4_ electrolyte concentration had the same effect on CAP degradation as current density. Acidic conditions (pH = 3) were clearly more favorable for CAP degradation. As the initial concentration of CAP increased (5→50 mg/L), the CAP degradation ratio and TOC removal rate gradually decreased, and the degradation amount increased.

When CAP is completely converted to inorganic substances such as CO_2_ and H_2_O before discharge into the environment, it has a minimal impact on ARG abundance in the aqueous environment and does not cause secondary pollution or pose environmental risks. However, when degradation wastewater containing incompletely degraded CAP and intermediates is discharged into the environment, it stimulates the expression of *intI1*, *cmlA*, *cmle3*, and *cata2*, leading to ARG enrichment in the environment.

## Figures and Tables

**Figure 1 ijerph-19-15632-f001:**
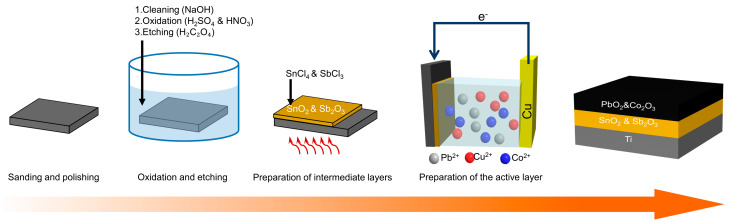
Ti/PbO_2_-Co electrode preparation process.

**Figure 2 ijerph-19-15632-f002:**
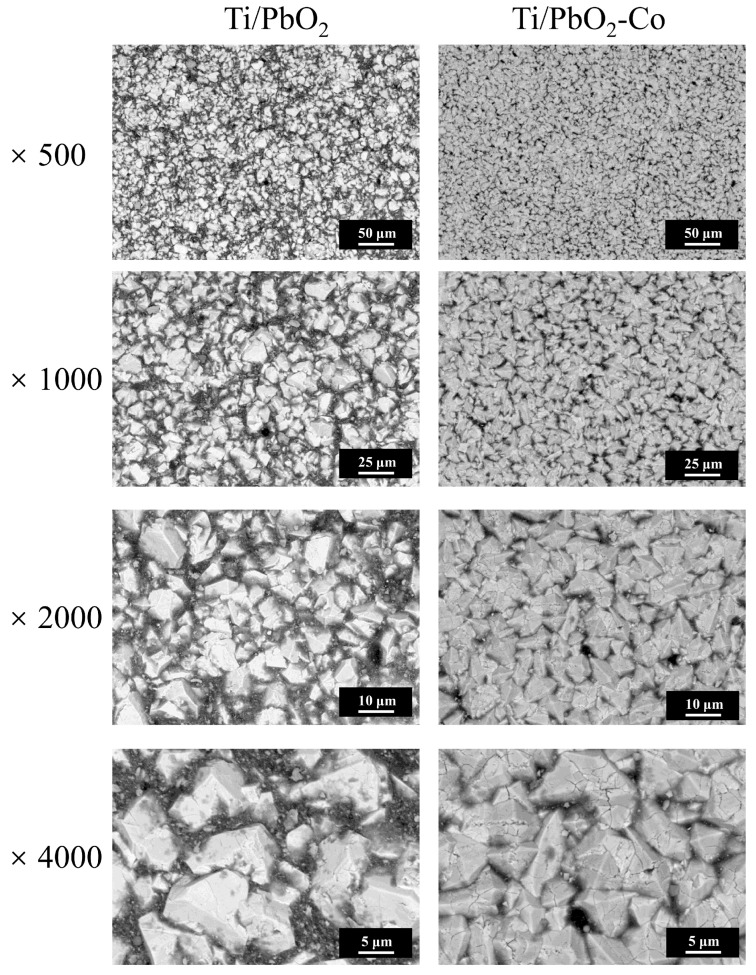
SEM images of the Ti/PbO_2_ surface.

**Figure 3 ijerph-19-15632-f003:**
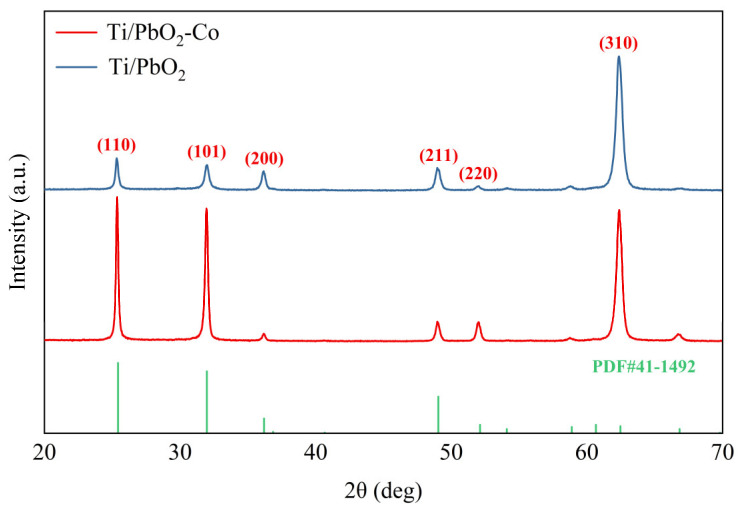
Results of the XRD analysis of Ti/PbO_2_ and Ti/PbO_2_-Co.

**Figure 4 ijerph-19-15632-f004:**
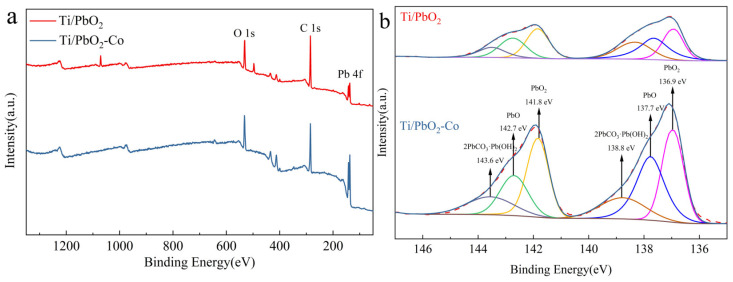
XPS image ((**a**): full spectrum; (**b**): Pb) of the Ti/PbO_2_ electrode.

**Figure 5 ijerph-19-15632-f005:**
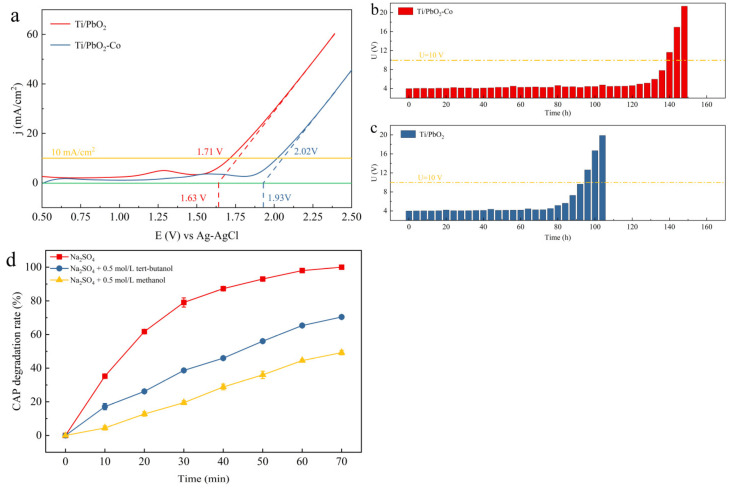
(**a**) LSV test of the Ti/PbO_2_ electrode; (**b**) accelerated life test for the Ti/PbO_2_ and (**c**) Ti/PbO_2_-Co electrodes; (**d**) free radical capture diagram.

**Figure 6 ijerph-19-15632-f006:**
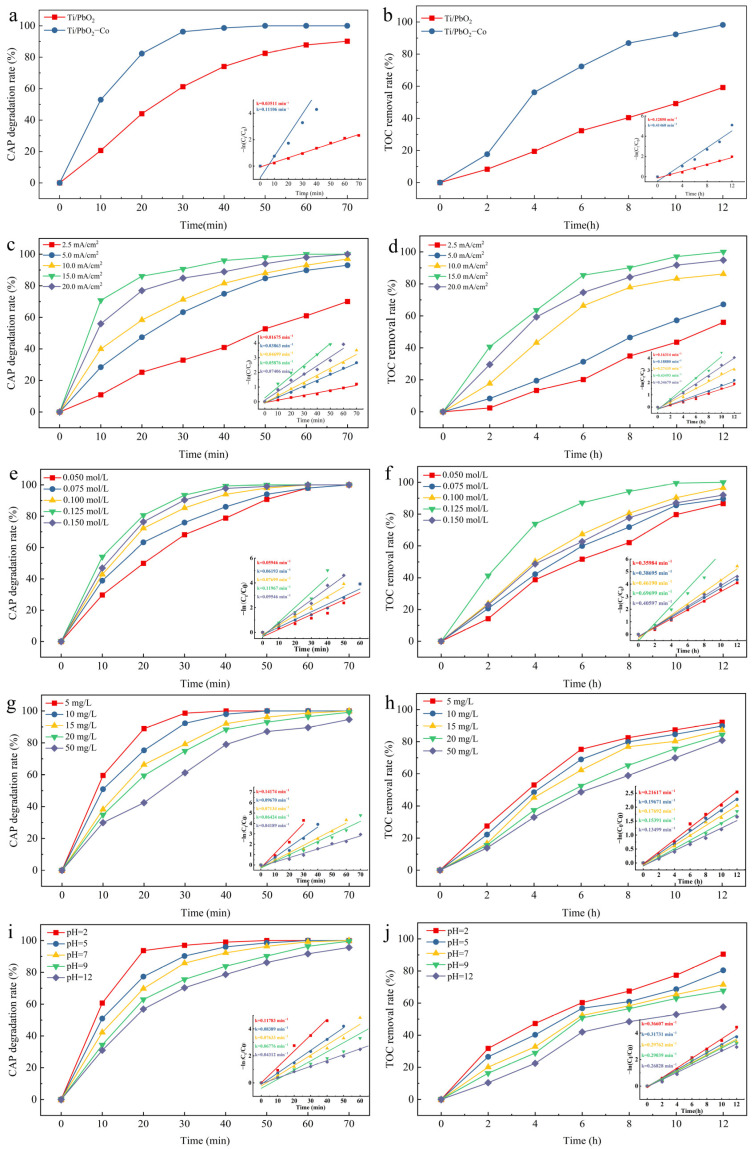
(**a**,**b**) Effect of Co elemental doping; (**c**,**d**) current density; (**e**,**f**) electrolyte concentration; (**g**,**h**) CAP concentration; and (**i**,**j**) solution pH on the CAP degradation and TOC removal rates.

**Figure 7 ijerph-19-15632-f007:**
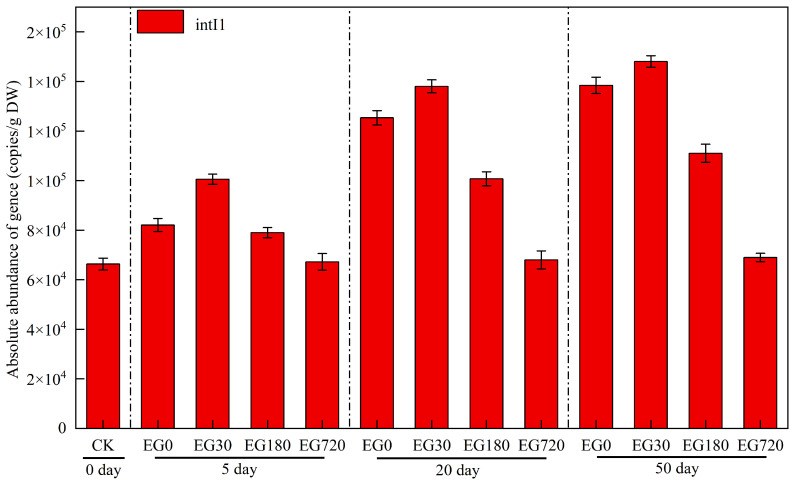
Changes in the absolute abundance of the *intI1* gene in various samples.

**Figure 8 ijerph-19-15632-f008:**
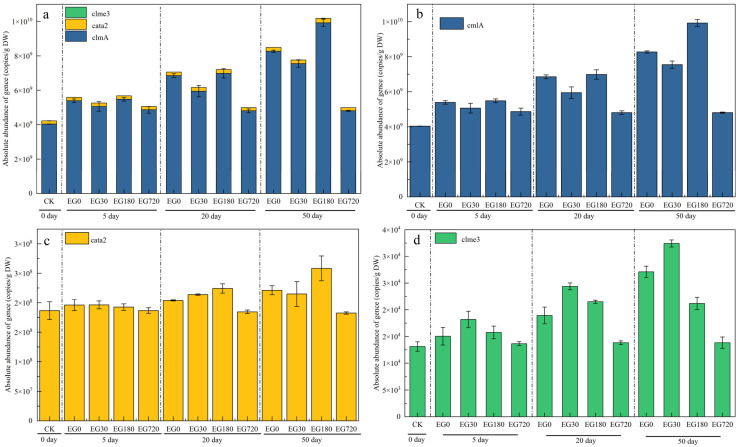
Changes in the absolute abundance of (**a**) the *cmlA*, *cmle3*, and *cata2* genes, (**b**) the *cmlA* gene alone, (**c**) the *cata2* gene alone, and (**d**) the *cmle3* gene alone.

## Data Availability

Not applicable.

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
