# Peer review of "Ti/PbO_2_ Electrode Efficiency in Catalytic Chloramphenicol Degradation and Its Effect on Antibiotic Resistance Genes"

_ijerph, 2022, doi:10.3390/ijerph192315632_

Round 1

Reviewer 1 Report

The manuscript entitled “Effectiveness of the Ti/PbO2 electrode catalytic degradation of chloramphenicol and its effect on antibiotic resistance genes” by Liu et al. prepared Ti/PbO2 electrodes using electrodeposition. Using Co doping, the Ti/PbO2 electrode exhibited improved electrochemical catalytic activity and lifetime when used in practice. The impact of different conditions on the effectiveness of the electrochemical degradation of chloramphenicol was investigated, and the most favorable conditions for chloramphenicol degradation were identified. They investigated the effects of the different stages of treatment with chloramphenicol solutions on the abundance of resistance genes in natural river substrates. When chloramphenicol was completely degraded (TOC removal = 100%), no effect on the resistance gene abundance in the river substrate was observed; the incomplete degradation of chloramphenicol significantly increased the absolute abundance of resistance genes. This suggests that when treating solutions with antibiotics, the complete degradation of antibiotics (TOC removal = 100%) is necessary before discharge into the environment to reduce secondary pollution. This study provides insights into the deep treatment of wastewater containing antibiotics and assesses the environmental impact of the resulting treated wastewater. However, I have some concerns and this work is publishable with these improvements.

1.      The abstract and conclusion should be rewritten in a scientific way.

2.      There are so many paragraphs in the introduction section. I suggest the author compose it into 4 paragraphs and remove the irrelevant sentences. The author also should clearly mention the aim of this work.

3.      The English of this manuscript should be highly improved. I suggest taking help from a native speaker.

4.      Figure 4; insets of the diagrams are not clear. I advise the author to make it visible to the audience/readers.

5.      There’re some valuable achievements in electrocatalysis recently, the authors may have a read for the further design of smart-structured electrocatalysts. Environ. Sci. Technol. 2013, 47, 10, 5353–5361; J. Phys. Chem. C 2021, 125, 23, 12707–12712; Journal of Industrial and Engineering Chemistry 105, 463-472; Mater. Chem. Front., 2022,6, 843-879; European Polymer Journal 157, 110635.

Author Response

Dear professor,

Thank you very much for reviewing my manuscript. For your questions, we have made careful and rigorous revisions. All revisions have marked in the manuscript. Below is our revised record.

Point 1: The abstract and conclusion should be rewritten in a scientific way.

Response 1: We have revised the abstract and conclusions. The incompleteness has been supplemented.

Point 2: There are so many paragraphs in the introduction section. I suggest the author compose it into 4 paragraphs and remove the irrelevant sentences. The author also should clearly mention the aim of this work.

Response 2: We have revised the introduction. The original 7 paragraphs have been revised to 5 paragraphs. Also removed irrelevant sections. This revision will make the introduction easier to focus on and more reader-friendly.

Point 3: The English of this manuscript should be highly improved. I suggest taking help from a native speaker.

Response 3: We have carefully revised the language of the manuscript.

Point 4: Figure 4; insets of the diagrams are not clear. I advise the author to make it visible to the audience/readers.

Response 4: We have resized Figure 4. This adjustment makes it easier for the audience/readers to see clearly.

Point 5: There’re some valuable achievements in electrocatalysis recently, the authors may have a read for the further design of smart-structured electrocatalysts. Environ. Sci. Technol. 2013, 47, 10, 5353–5361; J. Phys. Chem. C 2021, 125, 23, 12707–12712; Journal of Industrial and Engineering Chemistry 105, 463-472; Mater. Chem. Front., 2022,6, 843-879; European Polymer Journal 157, 110635.

Response 5: BES are a new field for obtaining better antibiotic treatment and reducing biotoxicity. The article (Accelerated Reduction of Chlorinated Nitroaromatic Antibiotic Chloramphenicol by Biocathode) gives us newer ideas for CAP treatment. We have therefore cited this excellent article in the article.

Again, the article (Recent advances in material design and reactor engineering for electrocatalytic ambient nitrogen fixation) gives us an expansion of ideas. It introduces us to another important application of electrochemical advanced oxidation technology. We quote from this article.

Reviewer 2 Report

Reviewer comments on IJERPH-2024495

Authors have developed Ti/PbO2 electrode and used for the electrochemical degradation of chloramphenicol and its effect on antibiotic resistance genes. The following points may be considered before acceptance: (Major revision).

1.     The same type of work has been published recently (Environmental Pollution Volume 301, 15 May 2022, 119031), what are the advantages of the present work compared with the published one? Although this paper was cited in the manuscript (ref 34), the novelty of the present work should be discussed.

2.     The preparation of Ti/PbO2 electrodes (section 2.3.1) is not clear

(i)              followed by the addition of 14.02 g of 142 tin chloride pentahydrate and 1.02 g of antimony chloride... why tin chloride?

(ii)            The gels were thermally baked 143 (T = 200°C, t = 30 min) and calcined (T = 550°C, t = 2.5 h) after the Ti plates were uniformly covered…how? What does it mean? When Ti plates were introduced into the precursor solution/mixture.

(iii)          The PbO2 deposition solution contained 165.6 g/L of Pb(NO3)2, 37.52 g/L of Cu(NO3)2, 0.42 g/L of NaF and 6.30 g/L of HNO3…. Why Cu(NO3)2 and NaF????

3.     In Figure 1. Ti/PbO2-CO should be Ti/PbO2-Co

4.     The oxidation state of each element presents in the Ti/PbO2-Co should be confirmed by XPS measurements. 

Author Response

Dear professor,

Thank you very much for reviewing my manuscript. For your questions, we have made careful and rigorous revisions. All revisions have marked in the manuscript. Below is our revised record.

Point 1: The same type of work has been published recently (Environmental Pollution Volume 301, 15 May 2022, 119031), what are the advantages of the present work compared with the published one? Although this paper was cited in the manuscript (ref 34), the novelty of the present work should be discussed.

Response 1: This article (Environmental Pollution Volume 301, 15 May 2022, 119031) is a pre-production of our subject group. Different types of experiments were carried out in parallel by our different learning groups.The article explores the effect of both Co and La doping on Ti/PbO2 electrodes, but lacks the effect of Co doping alone. Our manuscript focuses on the modification of Ti/PbO2 electrodes by doping with Co alone.

Point 2: The preparation of Ti/PbO2 electrodes (section 2.3.1) is not clear

(i)followed by the addition of 14.02 g of 142 tin chloride pentahydrate and 1.02 g of antimony chloride... why tin chloride?

(ii)The gels were thermally baked 143 (T = 200°C, t = 30 min) and calcined (T = 550°C, t = 2.5 h) after the Ti plates were uniformly covered…how? What does it mean? When Ti plates were introduced into the precursor solution/mixture.

(iii)The PbO2 deposition solution contained 165.6 g/L of Pb(NO3)2, 37.52 g/L of Cu(NO3)2, 0.42 g/L of NaF and 6.30 g/L of HNO3…. Why Cu(NO3)2 and NaF????

Response 2:

The process of Ti/PbO2-Co electrode preparation

(i)SnCl4 is a commonly used soluble salt to obtain Sn in water, and experiments can be carried out with SnCl4 in addition to other reagents that can be used to obtain Sn.

(ii) To prepare the SnO2-Sb2O3 intermediate layer, we first prepared a gel containing SnCl4 & SbCl3. The gel was first applied to the Ti substrate. The solvent is evaporated by baking and Sb is obtained at the same time. further by high temperature calcination, SnO2-Sb2O3 is immobilised on the Ti substrate.

(iii) Cu(NO3)2 is mainly in the reaction to provide Cu2+. Therefore, any reagent that can ionise Cu2+ in the reaction will do. NaF mainly provides F-, and F- can make the electrode surface denser and give a better performing electrode material.

Point 3: In Figure 1. Ti/PbO2-CO should be Ti/PbO2-Co

Response 3: In Figure 1, We have modified Ti/PbO2-CO to Ti/PbO2-Co and have made the substitution.

Point 4: The oxidation state of each element presents in the Ti/PbO2-Co should be confirmed by XPS measurements.

Response 4: We carried out XPS tests and added the results to the manuscript.

Reviewer 3 Report

In this work, Using Co dop-ing, the Ti/PbO2 electrode exhibited improved electrochemical catalytic activity and lifetime when used in practice. The impact of different conditions on the effectiveness of the electrochemical degradation of chloramphenicol was investigated, and the most favourable conditions for chloramphenicol degradation were identified. It's a comprehensive study, but some issues that need to be addressed during revision are listed below:

1. The introduction section should be a little more concise on a logical and clear basis, highlighting the focus of the study.

2. Some new TiO2 references should be added in the introduction such as Cell Reports Physical Science, 2022, 3, 101011; Journal of Catalysis, 401, 2021, 288-296.

3. There are many water treatment technologies, such as adsorption, photocatalysis and microwave catalysis, should be introduced in the introduction. The advantages and disadvantages of these techniques should be introduced. The new reviews and references on adsorption, photocatalytic purification and microwave catalysis of wastewater should be introduced and cited. Why select photocatalysis technology? Some new references should be added. Journal of Colloid and Interface Science, 2022, 625, 965-977; Journal of Catalysis, 406, 2022, 9-18

4. It is suggested to add the synthesis schematic of the material.

5. What are the advantages of the materials in this study compared to other catalysts of the same type?

6. XPS spectra for the typical sample of Ti/PbO2-Co and that after catalysis are suggested to be supplemented.

7. Lack of explanation of the degradation mechanism, which is crucial in this part.

8. There are some grammatical and formatting issues that need further correction.

Author Response

Dear professor,

Thank you very much for reviewing my manuscript. For your questions, we have made careful and rigorous revisions. All revisions have marked in the manuscript. Below is our revised record.

Point 1: The introduction section should be a little more concise on a logical and clear basis, highlighting the focus of the study.

Response 1: We have revised the introduction. The original 7 paragraphs have been revised to 5 paragraphs. Also removed irrelevant sections. This revision will make the introduction easier to focus on and more reader-friendly.

Point 2: Some new TiO2 references should be added in the introduction such as Cell Reports Physical Science, 2022, 3, 101011; Journal of Catalysis, 401, 2021, 288-296.

Response 2: We have studied the excellent performance of TiO2. The paper (RuO2/TiO2 photocatalysts prepared via a hydrothermal route: Influence of the presence of TiO2 on the reactivity of RuO2 in the artificial photosynthesis reaction) gives us a lot of help. We cite this article in the section on photocatalytic technology.

Point 3: There are many water treatment technologies, such as adsorption, photocatalysis and microwave catalysis, should be introduced in the introduction. The advantages and disadvantages of these techniques should be introduced. The new reviews and references on adsorption, photocatalytic purification and microwave catalysis of wastewater should be introduced and cited. Why select photocatalysis technology? Some new references should be added. Journal of Colloid and Interface Science, 2022, 625, 965-977; Journal of Catalysis, 406, 2022, 9-18

Response 4: We added the application of adsorption and photocatalytic oxidation in the field of antibiotic wastewater treatment. And we found that this paper (Dynamics of diffusion-limited photocatalytic degradation of dye by polymeric hydrogel with embedded TiO2 nanoparticles and Visible-light-assisted persulfate activation by SnS2/MIL-88B(Fe) Z-scheme heterojunction for enhanced degradation of ibuprofen) are of great help to our understanding of photocatalytic oxidation technology. We cited this excellent article.

Point 4: It is suggested to add the synthesis schematic of the material.

Response 4: We have drawn a flow chart of the Ti/PbO2-Co electrode preparation process.

Point 5: What are the advantages of the materials in this study compared to other catalysts of the same type?

Response 5: Carbon electrodes, as the earliest electrode material studied and industrially applied, have a strong adsorption capacity. However, they are prone to corrosion loss and contamination during the electrolysis process.

The monomers (Ni, Cu, Fe) are cheap, have good conductivity and are not easily contaminated, but can only be used at a limited applied potential and are prone to corrosion under over-acid conditions. The noble metals (Au, Ag, Lr) are surface stable, corrosion resistant and have good electrical conductivity, but they are expensive and cannot be used on a large scale. In addition to this, the electrode surfaces of the metal anodes are easily deactivated by passivation during the reaction process.

Ti/RuO2 electrodes have high electrocatalytic activity but poor stability, while IrO2 has high crystallinity but low electrical conductivity.

Ti/PbO2 electrodes have good properties for electrochemical catalytic reactions, and are chemically stable under strong acid and current conditions.

Point 6: XPS spectra for the typical sample of Ti/PbO2-Co and that after catalysis are suggested to be supplemented.

Response 6: We carried out XPS tests and added the results to the manuscript.

Point 7: There are some grammatical and formatting issues that need further correction.

Response 7: We have carefully revised the language of the manuscript.

Round 2

Reviewer 2 Report

 Accept in present form

Reviewer 3 Report

accepted